# Diagnosing and Treating Infertility via Transvaginal Natural Orifice Transluminal Endoscopic Surgery versus Laparoendoscopic Single-Site Surgery: A Retrospective Study

**DOI:** 10.3390/jcm12041576

**Published:** 2023-02-16

**Authors:** Yanli Zhang, Yiping Zhu, Mengsong Sui, Xiaoming Guan, Jing Sun

**Affiliations:** 1Shanghai Key Laboratory of Maternal Fetal Medicine, Shanghai Institute of Maternal-Fetal Medicine and Gynecologic Oncology, Shanghai First Maternity and Infant Hospital, School of Medicine, Tongji University, No. 2699, Gaoke West Road, Shanghai 200092, China; 2Division of Minimally Invasive Gynecology, Department of Obstetrics and Gynecology, Baylor College of Medicine, 6651 Main St., 10th Floor, Houston, TX 77030, USA

**Keywords:** transvaginal natural orifice transluminal endoscopic surgery, laparoendoscopic single-site surgery, infertility, hysterolaparoscopy

## Abstract

Objective: To evaluate the efficacy and safety of transvaginal natural orifice transluminal endoscopic surgery (vNOTES) in the treatment of female infertility. Materials and methods: This study includes 174 female patients with a history of long-standing female infertility. We retrospectively reviewed 41 patients who underwent hysterolaparoscopy (HL) via transvaginal natural orifice transluminal endoscopic surgery (vNOTES) and 133 patients who underwent laparoendoscopic single-site surgery (LESS). Demographic data, operation records, and pregnancy outcomes were collected and analyzed. The deadline for postoperative follow-up was June 2022. All the included patients were followed up for at least 18 months after surgery. Results: Compared with the LESS group, the vNOTES group had a shorter postoperative bowel movement time and less pain at 4 and 12 h (*p* = 0.004 vs. 0.008); no differences were found in other perioperative indicators. The clinical pregnancy rates of the vNOTES and LESS groups were 87.80% and 74.43% (*p* = 0.073), respectively. Conclusions: vNOTES represents a new, less invasive approach for infertility diagnosis and treatment that is particularly suitable for women who have special esthetic requirements. vNOTES is safe and practical and may be an ideal choice for scarless infertility surgery.

## 1. Introduction

Infertility is defined as the inability to conceive despite frequent unprotected sex for at least one year [1]. It causes immense psychoemotional disturbance among many couples and affects millions of families worldwide, involving approximately one in seven couples in developed countries and one in four couples in developing countries [2]. Hysterolaparoscopy (HL) is considered the gold standard for the evaluation of female infertility, particularly for detecting peritoneal endometriosis, adnexal adhesions, and uterine septa [3,4,5,6].

Advancements in minimally invasive technology and increased patient demand led to the development of natural orifice transluminal endoscopic surgery (NOTES), which represents one of the most important surgical innovations. NOTES can be performed via a variety of approaches, including through the stomach, esophagus, bladder, and rectum, but it is performed transvaginally in the majority of cases.

In its early days, transvaginal NOTES (vNOTES) was only used for diagnostic purposes or performed with transabdominal assistance [7,8,9]. The first published instance was a transvaginal endoscopic cholecystectomy performed by Zorron et al. at the University Hospital of Teresopolis, Brazil [10]. Lee et al. reported the first case series of pure vNOTES procedures for adnexal diseases in 2012 [11]. The acceptance of operations performed using vNOTES is rising, with the expectation that it will soon become established as an alternative technique to traditional laparoscopy.

Previous studies have not investigated whether vNOTES or LESS is superior in treating cases of infertility. The objective of this study is to demonstrate the feasibility, safety, and efficacy of vNOTES in the surgical treatment of female infertility overall and in comparison with LESS.

## 2. Materials and Methods

### 2.1. Ethical Approval

This study was conducted in accordance with the Declaration of Helsinki; the protocol was approved by the Ethics Committee of Shanghai First Maternity and Infant Hospital (project identification code No. KS1963). The date of approval was June 2019. Obstetrical data were collected via telephone interview.

### 2.2. Subject Selection

This retrospective single-center study was conducted at Shanghai First Maternity and Infant Hospital, School of Medicine, Tongji University, from September 2017 to June 2022. We identified 193 patients with a diagnosis of female infertility who underwent HL for the first time in our hospital.

The inclusion criteria were as follows: (1) unable to conceive despite frequent unprotected sex for at least one year and (2) age at surgery between 20 and 45 years old. The exclusion criteria were as follows: (1) male infertility; (2) no fertility intention; (3) severe systemic disease; (4) other gynecological condition that may cause infertility, including amenorrhea and premature ovarian failure (POF); and (5) malignant tumor.

After case selection according to the inclusion and exclusion criteria, 174 patients were included in the study. The exclusion criteria resulted in the removal of 19 patients due to personal reasons (n = 7), insufficient follow-up data (n = 7), fibroids (n = 3), diabetes (n = 1), and malignant tumor (n = 1).

The deadline for postoperative follow-up was June 2022. All included patients were followed up with for at least 18 months after surgery.

### 2.3. Clinical Definitions

The term primary infertility was used to indicate a couple without a pregnancy history, while secondary infertility was used to designate couples who had experienced at least one conception. Clinical pregnancy was defined as the presence of at least one gestational sac on ultrasound. Biochemical pregnancies were not included due to the limitations of the retrospective study design. Spontaneous miscarriage was defined as spontaneous loss of a fetus before 28 weeks of pregnancy, in accordance with Chinese expert consensus [12]. Pregnancy rate was defined as the proportion of pregnant patients to the total number of patients. Spontaneous miscarriage rate was defined as the proportion of spontaneous miscarriages to total pregnancies (induced abortion numbers were not included). Live birth rate was defined as the proportion of live births to total pregnancies, with ongoing pregnancies excluded.

### 2.4. Surgical Intervention, Variables, and Measurements

Ultrasonographies were performed during the proliferative phase of the menstrual cycle (days 3–7 after menses phase). All the patients signed informed-consent forms prior to surgery. Presurgical symptoms, surgical data, and postsurgical symptoms were retrieved from patient admission and operative databases. Pregnancy outcomes were collected via telephone interview. In this study, independent variables were classified into three groups: demographic factors, including age, body mass index (BMI), type of infertility (primary infertility or secondary infertility), infertility duration, abortion history, gravida, parity, and history of surgery; surgical factors, including drop in Hb (hemoglobin) and Hct (hematocrit), blood loss, surgical time, anal exhaust time, postoperative pain NRS (numerical rating scale), postoperative fever, postoperative hospital stay, and TTP (time to pregnancy); and pregnancy outcomes, including mode of conception (spontaneous conception or ART), live birth, spontaneous miscarriage, ongoing pregnancy, preterm labor, and spontaneous conception.

NRS is an 11-point (NRS-11) scale which evaluates pain and is wildly used in clinical settings because it is easy to administer and score [13]. NRS: the patient is asked to indicate the value of their pain on the scale. An 11-point scale was used, with “0” representing “no pain” and “10” representing the “most severe pain imaginable,” at the time of assessment. The mean blood loss during surgery was estimated by weighing and area methods. Postoperative fever is usually defined as any oral temperature of 38.0 °C or more occurring 24 or more hours postoperatively [14].

We used clinical pregnancy as the primary outcome, which is defined as the presence of at least one gestational sac on ultrasound.

### 2.5. Surgical Procedures

The vNOTES surgical procedure is briefly described as follows:After administration of general anesthesia and endotracheal intubation, patients were placed in the Trendelenburg position;Each vNOTES operation started with conventional vaginal surgery by creating a 2 cm posterior colpotomy;A small wound retractor (HTKD-Med, HK-120/100-60/70D, Beijing, China) was inserted and fixed from behind the colpotomy to the vaginal introitus to maximize and protect the incision area (Figure 1A). Then, a port was established in the vagina using two 3–5 mm and two 5–12 mm cannulas, as used in transumbilical single-port laparoscopy (Figure 1B);Instruments used in all vNOTES procedures included a 30° endoscope (KARL STORZ GmbH & Co., KG., Tuttlingen Germany), conventional rigid laparoscopic instruments, and a 6.5 mm, 22° endoscope (HAWK GMBH, W5050, Hangzhou, China) (Figure 1C,D).

The surgical procedures are similar for LESS and traditional hysteroscopy and laparoscopy.

### 2.6. Statistical Analysis

SPSS 25.0 software (IBM SPSS, Chicago, IL, USA) was used for data analysis. The graphical method is effective for testing the normality of distribution. Continuous variables such as age and BMI are presented as means ± SDs, whereas discrete variables such as gravidity and parity are presented as median values and ranges. Comparisons between two groups were performed using unpaired two-tailed Student’s t-test for normally distributed parameters. For data that were not normally distributed, non-parametric methods were used for data analysis. Categorical variables were compared using χ^2^ test. Statistical differences were considered significant when *p* < 0.05.

## 3. Results

### 3.1. Patient Characteristics

A total of 174 (90.16%) patients with infertility underwent HL, which included 41 infertile women in the vNOTES group and 133 women in the LESS group. The baseline characteristics of the included subjects are summarized in Table 1. No statistically significant difference (*p* = 0.239) was found in the age of the two groups (30.93 ± 3.47 years vs. 31.80 ± 4.30 years). There were no significant differences in demographic characteristics between the two groups.

### 3.2. Surgical Findings

Surgeries using the standard method are reported in Table 2. A combination of hysteroscopy and laparoscopy in one vNOTES patient is included (Figure 2).

There were no significant difference in mean blood loss between the vNOTES and LESS groups (20 (5~50) mL vs. 20 (10~50) mL, *p* = 0.933), which were ≤50 mL. The duration of surgery were slightly longer for the vNOTES group than the LESS group but not statistically significantly different (86.76 ± 35.22 min vs. 83.54 ± 34.80 min, *p* = 0.607). There were no significant differences between the two groups regarding postoperative fever rates (39.02% vs. 45.86%, *p* = 0.444) or postoperative hospital stay (3.46 ± 0.126 d vs. 3.64 ± 0.073 d, *p* = 0.238). No significant differences between the groups were found in terms of postoperative hemoglobin (Hb) drop (10.58 ± 8.27 g/L vs. 11.05 ± 6.85 g/L, *p* = 0.717) and hematocrit (Hct) drop (4.15 ± 9.36% vs. 3.05 ± 2.10%, *p* = 0.456). There were significant difference in anal exhaust time between the two groups (*p* = 0.009), while there were no significant difference in postoperative pregnancy time (*p* = 0.195) (Table 3). No intraoperative or postoperative complications such as severe bleeding, rectal injury, or severe infection occurred in either group. 

### 3.3. Fertility Results and Pregnancy Outcomes

At long-term follow-up (median 37 months, range 18–57 months), 26 (14.94%) patients had failed to conceive. A total of 13 (7.47%) patients had stopped trying to conceive, including two due to premature ovarian failure (POF), four due to age > 46.5 years, and seven due to repeated IVF failure (Figure 3). The clinical pregnancy rates in the vNOTES and LESS groups were 87.80% and 74.43%, respectively (*p* = 0.073). Postoperative reproductive outcomes were also compared between the two groups (Table 4).

## 4. Discussion

Following the emergence of novel technologies, increasing attention has been paid to vNOTES, which represents a major paradigm shift in scarless surgery [15,16]. vNOTES surgery has become a novel type of gynecological micro-non-invasive technology which is prevalent all over the world. To the best of our knowledge, this is the first study to compare vNOTES and LESS for patients with infertility. Given the current interest in vNOTES, this study reports its innovative use in the diagnosis and treatment of infertility. We, further, compared the feasibility and surgical outcomes of vNOTES with LESS. Compared with LESS and traditional laparoscopy, vNOTES is associated with significantly lower pain scores and a shorter postoperative hospital stay. An overwhelming body of evidence in the literature indicates that minimally invasive surgery not only offers superior cosmetic results but—most importantly—also reduces surgical trauma, blood loss, inflammatory response, postoperative pain, and recovery time [11,17,18,19,20,21,22]. There was no significant difference in bleeding, duration of surgery, or drop in Hb or Hct between the two groups. The durations of surgery were slightly longer for the vNOTES group than the LESS group, which were consistent with the research of Hou et al. [23].

In addition, 36 vNOTES patients (87.8%) and 84 LESS patients (63.2%) had an event of anal exhaust within 12 h after the operation, which are significantly different (*p* = 0.002). Anal-exhaust time is an important observation index for early postoperative intestinal function recovery [23,24].

We conclude that vNOTES is superior to LESS in terms of allowing a faster return to intestinal function, based on anal-exhaust time, and normal activities. The results of the present study demonstrate that the novel method described here is technically practical and safe, with a reasonable operation time and blood loss and a rapid recovery time. Baekelandt et al. [25] and Sowmya et al. [26] suggested that vNOTES might allow more women to be treated in an outpatient setting.

It is well-known that early postoperative pain levels are related to incision pain [27]. Differences in umbilical and vaginal innervations mainly affect early postoperative pain levels [28]. vNOTES was found to be associated with lower pain scores at 4 h after surgery [19,29,30,31], which is consistent with the results of our study.

A transvaginal approach of introducing surgical instruments through nonsterile orifices into the normally sterile peritoneal cavity has the potential to lead to adverse infection events. A previous study [32] found that most gynecologists are concerned about the potential risk of infection, whereas in our study, we find that vNOTES does not increase the risk of postoperative infection (39.02% vs. 45.86%, *p* = 0.444). These infection rates appear to be comparable with those of a transabdominal approach, and there is no strong evidence from bacteriologic studies to support any fears regarding infection [33]. In this study, no adverse infection events were found in any patients.

The clinical pregnancy rate was slightly higher in the vNOTES group than in the LESS group; however, there was no significant difference between the two groups (87.8% and 70.21%, respectively, *p =* 0.073). Ekine et al. reported an overall pregnancy rate of 81.3% [34], while the total pregnancy rate after vNOTES in this study was 87.8%. Feng et al. [35] showed that vNOTES surgery does not affect subsequent pregnancy and delivery.

The efficacy and safety of the transvaginal route has been established; as a result of this study, we conclude that vNOTES is a practical and safe surgical technique for performing infertility procedures and shows similar surgical outcomes and superior cosmesis compared with LESS [36].

Our data show that, if performed by experienced surgeons, vNOTES offers less early postoperative pain and a shorter time to anal exhaust than LESS. To date, the field of minimally invasive surgery has experienced enormous innovation, and future studies are focusing on decreasing abdominal pain, minimizing infection risk, reducing hernia incidence, and minimizing any negative cosmetic effects associated with the surgery [37].

Compared with LESS, vNOTES offers a larger space and a decreased incidence of instrument clashing during handling because of the large colpotomy incision [38,39]. A reduction in port-associated complications such as hernias, vascular and soft-tissue traumatic injuries during trocar insertion, and nerve injuries was also demonstrated. The operation radius of transvaginal laparoscopic surgery is small, which undoubtedly reduces the difficulty of the operation. Additionally, in this approach, abdominal-wall vessel injury associated with trocar insertion is avoided. vNOTES can be used to detect and treat various structural abnormalities in multiple sites—such as the pelvis, fallopian tubes, and uterus—while, at the same time, using the same transvaginal route and may be recommended as the first and final procedure for the evaluation of female infertility. It offers a new vision of infertility surgery, accompanied by a more convenient and safer procedure, completely avoiding the development of wounds and scars on the abdominal wall. All patients were satisfied with the scar-free cosmetic results of this surgical procedure.

This study has some limitations, too. Firstly, the main limitation of this study is the lack of randomization due to its retrospective design. Secondly, when interviewed by telephone, subjects were provided with the purpose of the study, and they were able to refuse to participate in the study without giving any reason. In addition, telephone interview may have led to a loss of follow-up and recall errors. Further prospective studies with larger sample sizes are required to draw firmer conclusions.

When performed by experienced surgeons and with the proper selection of patients, transvaginal hysterolaparoscopy can be an appropriate and practical same-day procedure for the evaluation of female infertility. This is the first IDEAL (idea, development, exploration, assessment, long-term) study that confirms the feasibility of vNOTES for infertility, although further investigations are required.

With ongoing efforts to decrease the size and/or number of incisions, improve patient outcomes, and increase patient satisfaction, vNOTES for infertility represents the next leap in laparoscopy.

## 5. Conclusions

vNOTES is a novel and revolutionary surgical technique in the field of minimally invasive surgery. It can be performed securely and effectively in infertility patients and offers a new, safe, and scarless way to diagnose and treat female infertility. In future, further comparative studies or even prospectively randomized controlled trials should be performed to confirm the advantages and significance of vNOTES in infertility patients.

## Figures and Tables

**Figure 1 jcm-12-01576-f001:**
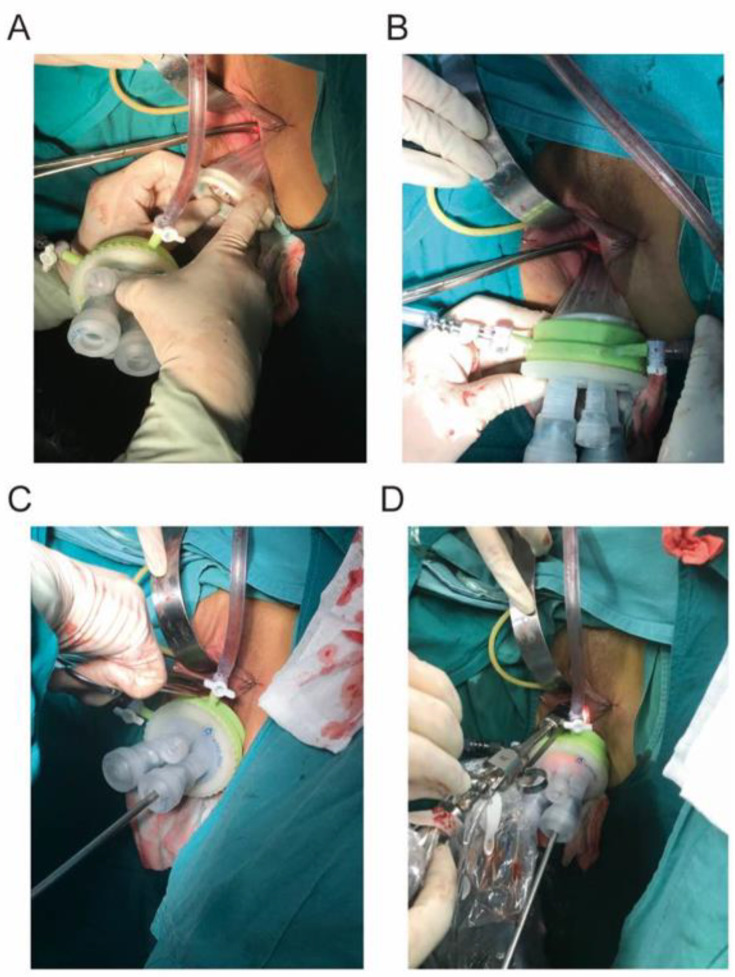
The wound retractor is inserted and fixed from behind the colpotomy to the vaginal introitus to maximize and protect the incision area (**A**). Establishment of the port (**B**). vNOTES surgical access is established (**C**,**D**).

**Figure 2 jcm-12-01576-f002:**
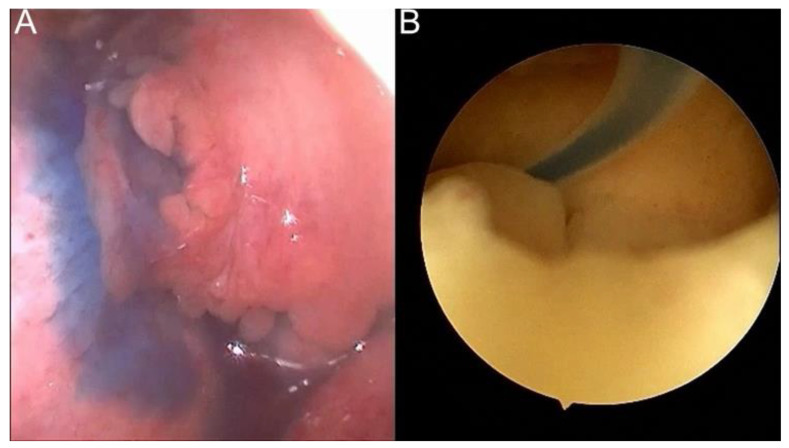
Combination of hysteroscopy and laparoscopy in one vNOTES patient. (**A**) laparoscopic salpingotomy; (**B**) hysteroscopic salpingostomy.

**Figure 3 jcm-12-01576-f003:**
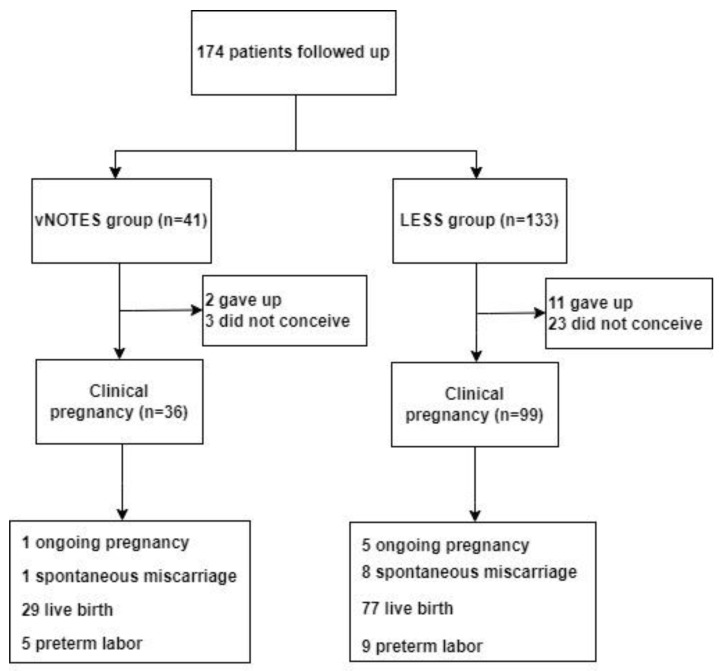
Flowchart of reproductive outcomes in both groups.

**Table 1 jcm-12-01576-t001:** Comparison of preoperative demographic and clinical characteristics.

Characteristics	^b^ vNOTES Group (n = 41)	^c^ LESS Group (n = 133)	*p* Value
Age (yrs)	30.93 ± 3.47	31.80 ± 4.30	0.239
^a^ BMI (kg/m^2^)	20.98 ± 2.35	21.80 ± 3.63	0.093
Type of infertility n (%)			0.342
Primary infertility	28 (68.29%)	80 (60.15%)	
Secondary infertility	13 (31.71%)	53 (39.85%)	
Abortion history n (%)	13 (31.71%)	42 (31.58%)	0.988
Infertility duration (months)	18 (12~30)	24 (12~36)	0.370
Gravida	0 (0~1)	0 (0~1)	0.679
Parity	0 (0~1)	0 (0~1)	0.561
History of surgery n (%)	9 (21.95%)	49 (36.84%)	0.059

Notes: ^a^ BMI, body mass index; ^b^ vNOTES: transvaginal natural orifice transluminal endoscopic surgery; ^c^ LESS: laparoendoscopic single-site surgery; data are presented as means ± SDs, medians (interquartile ranges), or n (%).

**Table 2 jcm-12-01576-t002:** Minimally invasive surgeries for infertility: vNOTES and LESS groups.

	^c^ vNOTES Group (n = 41)	^d^ LESS Group (n = 133)
^a^ Laparoscopy	3 (7.32%)	8 (6.02%)
Laparoscopic adnexal cyst surgery	15 (36.59%)	15 (11.28%)
Laparoscopic salpingotomy	16 (39.02%)	66 (49.62%)
Unilateral	10	24
Bilateral	6	42
Laparoscopic salpingoplasty	13 (31.71%)	25 (18.80%)
Subserous myoma	2 (48.78%)	16 (12.03)
^b^ Hysteroscopy	1 (24.39%)	12 (9.02%)
Intrauterine adhesions (IUA)	20 (48.78%)	71 (53.38%)
^e^ IUD	4 (9.76%)	26 (19.55%)
Salpingostomy	32 (78.05%)	82 (61.65%)
Endometrial polyps	11 (26.83%)	40 (30.08%)
Uterine septum	4 (9.76%)	0

Notes: ^a^ laparoscopic surgery; ^b^ hysteroscopic surgery; ^c^ vNOTES: transvaginal natural orifice transluminal endoscopic surgery; ^d^ LESS: laparoendoscopic single-site surgery; ^e^ IUD: intrauterine device.

**Table 3 jcm-12-01576-t003:** Surgical outcomes of vNOTES and LESS groups (n = 174).

Category	^e^ vNOTES Group (n = 41)	^f^ LESS Group (n = 133)	*p* Value
Drop in Hb ^a^ (g/L)	10.58 ± 8.27	11.05 ± 6.85	0.717
Drop in Hct ^b^ (%)	4.15 ± 9.36	3.05 ± 2.10	0.456
Intraoperative blood loss (mL)	20 (5~50)	20 (10~50)	0.933
Operation duration (min)	86.76 ± 35.22	83.54 ± 34.80	0.607
Anal exhaust time			0.002 **
≤12 h	36 (87.80%)	84 (63.16%)	
≥12 h	5 (12.20)	49 (36.84%)	
Postoperative pain NRS ^c^			
4 h	2.07 ± 0.93	2.56 ± 0.80	0.004 **
12 h	2.46 ± 0.67	2.79 ± 0.64	0.008 **
Postoperative fever (%)	16 (39.02%)	61 (45.86%)	0.444
Postoperative hospital stay (d)	3.46 ± 0.126	3.64 ± 0.073	0.238
TTP ^d^	36 (87.8%)	99 (74.43%)	0.195
<6 months	11 (30.56%)	48 (48.48%)	
6–12 months	14 (38.89%)	23 (23.23%)	
12–24 months	9 (25%)	22 (22.22%)	
>24 months	2 (5.56%)	6 (6.07%)	

Notes: ^a^ Hb: hemoglobin; ^b^ Hct: hematocrit; ^c^ NRS: numerical rating scale; ^d^ TTP: time to pregnancy; ^e^ vNOTES group: transvaginal natural orifice transluminal endoscopic surgery group; ^f^ LESS group: laparoendoscopic single-site surgery group. NRS was performed at 4 and 12 h after operation. ** *p* < 0.01.

**Table 4 jcm-12-01576-t004:** Comparison of postoperative reproductive outcomes between two groups.

	^b^ vNOTES Group (n = 41)	^c^ LESS Group (n = 133)	*p* Value
Pregnant patients	36 (87.80%)	99 (70.21%)	0.073
Mode of conception			0.098
Spontaneous conception	20 (48.78%)	45 (33.83%)	
^a^ ART	21 (51.22%)	88 (66.17%)	
Pregnancy outcome			0.645
Live birth	29 (80.55%)	77 (77.78%)	
Spontaneous miscarriage	1 (2.78%)	8 (8.08%)	
Ongoing pregnancy	1 (2.78%)	5 (5.05%)	
Preterm labor	5 (13.89%)	9 (9.09%)	
Conception mode of live birth			0.696
Spontaneous conception	19 (52.78%)	57 (57.58%)	
ART	17 (47.22%)	42 (42.42%)	

Notes: ^a^ ART: assisted reproductive technology; ^b^ vNOTES group: transvaginal natural orifice transluminal endoscopic surgery group; ^c^ LESS group: laparoendoscopic single-site surgery group.

## Data Availability

The data that support the findings of this study are available on reasonable request to the corresponding author. The data are not publicly available due to privacy or ethical restrictions.

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
