# Peer review of "Diagnosing and Treating Infertility via Transvaginal Natural Orifice Transluminal Endoscopic Surgery versus Laparoendoscopic Single-Site Surgery: A Retrospective Study"

_jcm, 2023, doi:10.3390/jcm12041576_

Round 1
Reviewer 1 Report (Previous Reviewer 1)
This is a new submission which the authors evaluated the efficacy and safety of natural orifice transvaginal transluminal endoscopic surgery in the treatment of female infertility. They concluded that transvaginal transluminal endoscopic surgery offers a new, less invasive approach to the diagnosis and treatment of infertility that is particularly suitable for women with special aesthetic needs. They also pointed out that natural orifice transvaginal transluminal endoscopic surgery is safe and feasible, which may be an ideal choice for scarless infertility surgery.
The revised version has been significantly improved. The authors have incorporated all my previous comments into the revised version.
However, some minor corrections are still needed:
1. Please provide the exact primary and secondary outcomes of the study. Paragraph (clinical definitions are all well and good, but do not represent the results of the study). Please revise.
2. Tables – There are few abbreviations that were not mentioned in a legend of the Tables. Please revise.
Author Response
Response to Reviewer 1 Comments
Point 1: Please provide the exact primary and secondary outcomes of the study. Paragraph (clinical definitions are all well and good, but do not represent the results of the study). Please revise.
Response 1: Thank you very much indeed for your comments. We used clinical pregnancy as the primary outcome, which is defined as the presence of at least one gestational sac on ultrasound. Please see the lines 112-113 in the Manuscript.
Point 2: Tables – There are few abbreviations that were not mentioned in a legend of the Tables. Please revise.
Response 2: Thank you very much indeed for your comments. We have revised the abbreviations that were mentioned in a legend of the Tables. Please see the Manuscript for details.

Reviewer 2 Report (Previous Reviewer 3)
The authors made some of the suggested changes which improved certain sections of the manuscript. Manuscript also contains several pictures which are quite nice and informative for the readers, particularly those not familiar with the vNOTES surgical technique. Still, there are several important methodological and clinical issues that have not been solved in the resubmitted manuscript. Spelling and grammar errors were not corrected throughout the manuscript (i.e., “hyeterscopy”-page 3).
Furthermore, some of the performed changes impair the quality of the presented work: i.e., third and fourth line of the abstract contain repetition of the same text.
The Materials and Methods section still needs substantial clarification. Verbal informed consent is not common way to obtain patient’s approval.
Clinical pregnancy rate can be defined only as a pregnancy in which clinical signs of the fetus can be either seen or heard. This type of pregnancy can be confirmed through ultrasound visualization of the gestational sac or heartbeat. Positive urine/blood pregnancy test implies only biochemical pregnancy, which may not become a clinical pregnancy. Thus, authors must distinguish between these two categories.
The authors did not explain in the M&M section which method was used to estimate mean blood loss during the surgery and which criterion was used to define postoperative fever.
The authors claim that mean blood loss represents a continuous variable and follows the normal distribution. It does not seem so, judging from the values presented in their work (it is expressed by median±SD and the SD is greater than mean value). There are numerous issues considering statistical methodology and results in the presented work. It is not proper to compare data presented by median and range using Student’s test, as such data should be compared using non-parametric methods. The use of such statistical tests was not mentioned in the M&M section.
The sentence “All the operations were performed by a surgical team.” is unclear. Being a surgeon does not define surgical experience, and all the surgeries are performed by surgeons. There are, of course some exceptions in the underdeveloped world where cesarean sections are performed by GPs, LLETZ procedures performed by nurses, etc. but the procedures elaborated in the presented manuscript are too complex to be performed in such a way.
The authors did not provide plausible explanation for using the IQR instead of range in Table 1. Results for infertility duration are close to the level of statistical significance-the authors should comment this in the text.
Regarding Table 2. title should be more informative than it is.
Sentence “vNOTES group significant difference in both groups (10.58±8.27 g/l vs ... “is completely unclear. This sentence needs clarification. The authors do not mention Table 3. in the manuscript body.
In the section “Fertility results and pregnancy outcomes” of the manuscript body authors claim that a total of 26 patients failed to conceive and 13 stopped trying to conceive, as shown in figure 3. On the contrary, in the box on the right side of Figure 3 numbers are as follows: 23 did not conceive and 11 give up. Which numbers are correct?
Drawing conclusions on surgical experience as a factor of safety and feasibility of vNOTES is not supported by the presented results. There is no data regarding postoperative infection definition criteria, although authors discuss postoperative infections rate. Judging from Table 3. percentages which are discussed are related to postoperative fever. How did the authors exclude use of electrosurgery as a cause of increased temperature? Postoperative fever cannot be considered as a sure sign of postoperative infection.
References shoul be properly cited, i.e. Fertil Steril, etc.
This paper could be improved according to reviewer comments.
Author Response
Response to Reviewer 2 Comments
Point 1: The authors made some of the suggested changes which improved certain sections of the manuscript. Manuscript also contains several pictures which are quite nice and informative for the readers, particularly those not familiar with the vNOTES surgical technique. Still, there are several important methodological and clinical issues that have not been solved in the resubmitted manuscript. Spelling and grammar errors were not corrected throughout the manuscript (i.e., “hyeterscopy”-page 3).
Response 1: Thank you very much indeed for your comments. We have revised the spelling and grammar errors throughout the manuscript and the English editing service is finished.
Point 2: Furthermore, some of the performed changes impair the quality of the presented work: i.e., third and fourth line of the abstract contain repetition of the same text.
Response 2: Thank you very much indeed for your comments. We have used the English editing services of MDPI and the English editing service is finished.
Point 3: The Materials and Methods section still needs substantial clarification. Verbal informed consent is not common way to obtain patient’s approval.
Response 3: Thank you very much indeed for your comments. We have clarified the Materials and Methods section in the manuscript. Obstetrical data were collected via telephone interview.
Point 4: Clinical pregnancy rate can be defined only as a pregnancy in which clinical signs of the fetus can be either seen or heard. This type of pregnancy can be confirmed through ultrasound visualization of the gestational sac or heartbeat. Positive urine/blood pregnancy test implies only biochemical pregnancy, which may not become a clinical pregnancy. Thus, authors must distinguish between these two categories.
Response 4:Thank you very much indeed for your comments. Clinical pregnancy was defined as amenorrhea, positive urine/blood pregnancy test, and intrauterine pregnancy diagnosed by ultrasonography. We used clinical pregnancy as the primary outcome, which is defined as the presence of at least one gestational sac on ultrasound. Biochemical pregnancies weren’t included due to the limitation of the retrospective study design. Please see the lines 112~113 in the manuscript.
Point 5: The authors did not explain in the M&M section which method was used to estimate mean blood loss during the surgery and which criterion was used to define postoperative fever.
Response 5:Thank you very much indeed for your comments. The mean blood loss during the surgery was estimated by weighing and area methods. Postoperative fever is usually defined as a temperature > 38 ° C within 48 hours after surgery. Please see the lines 113~115 in the manuscript.
Point 6: The authors claim that mean blood loss represents a continuous variable and follows the normal distribution. It does not seem so, judging from the values presented in their work (it is expressed by median±SD and the SD is greater than mean value). There are numerous issues considering statistical methodology and results in the presented work. It is not proper to compare data presented by median and range using Student’s test, as such data should be compared using non-parametric methods. The use of such statistical tests was not mentioned in the M&M section.
Response 6:Thank you very much indeed for your comments. We re-analyzed the data. The mean blood loss are not normally distributed, non-parametric methods were used for data analysis. And such statistical tests was mentioned in the M&M section.
Point 7: The sentence “All the operations were performed by a surgical team.” is unclear. Being a surgeon does not define surgical experience, and all the surgeries are performed by surgeons. There are, of course some exceptions in the underdeveloped world where cesarean sections are performed by GPs, LLETZ procedures performed by nurses, etc. but the procedures elaborated in the presented manuscript are too complex to be performed in such a way.
Response 7:Thank you very much indeed for your comments. All surgeries were performed by the same surgical team.
Point 8: The authors did not provide plausible explanation for using the IQR instead of range in Table 1. Results for infertility duration are close to the level of statistical significance-the authors should comment this in the text.
Response 8:Thank you very much indeed for your comments. We used the IQR in Table 1 for Infertility duration, Gravida and Parity. Please refer to Table 1.
Point 9: Regarding Table 2. title should be more informative than it is.
Response 9:Thank you very much indeed for your comments. We have revised the title of Table 2. Please see the manuscript for details.
Point 10: Sentence “vNOTES group significant difference in both groups (10.58±8.27 g/l vs ... “is completely unclear. This sentence needs clarification. The authors do not mention Table 3. in the manuscript body.
Response 10:Thank you very much indeed for your comments. We had revised the sentence. Please see the manuscript for details.
Point 11: In the section “Fertility results and pregnancy outcomes” of the manuscript body authors claim that a total of 26 patients failed to conceive and 13 stopped trying to conceive, as shown in figure 3. On the contrary, in the box on the right side of Figure 3 numbers are as follows: 23 did not conceive and 11 give up. Which numbers are correct?
Response 11:Thank you very much indeed for your comments. Both numbers are correct. 26 (14.94%) patients failed to conceive. 13 (7.47%) patients stopped trying to conceive including 2 due to premature ovarian failure (POF), 4 due to age>46.5 years and 7 due to repeated IVF failures
Point 12: Drawing conclusions on surgical experience as a factor of safety and feasibility of vNOTES is not supported by the presented results. There is no data regarding postoperative infection definition criteria, although authors discuss postoperative infections rate. Judging from Table 3. percentages which are discussed are related to postoperative fever. How did the authors exclude use of electrosurgery as a cause of increased temperature? Postoperative fever cannot be considered as a sure sign of postoperative infection.
Response 12:Thank you very much indeed for your comments. Postoperative fever is usually defined as a temperature > 38 ° C within 48 hours after surgery. The main limitation of this study is the lack of randomization due to its retrospective design. Further prospective studies with larger sample sizes are required to draw firmer conclusions.
Point 13: References should be properly cited, i.e. Fertil Steril, etc.
Response 13: We had properly cited the references. Thank you very much indeed for your comments.

Reviewer 3 Report (Previous Reviewer 2)
I have a few comments below:
In the Introduction section, the authors have explained the abbreviations (vNOTES and LESS) used many times. Usually, abbreviations are explained only once.
In the Results section the authors give the number of pregnancies obtained; there is no information on whether these were natural pregnancies or those obtained through assisted reproduction techniques; if so, which one?
The authors still did not include information on how many points on the pain scale (NRS) patients gave after using both methods.
If the authors compare the duration of pain, they should give at least a cut-off point.
In my opinion, if we compare pain when using some methods, then data (in the form of NRS scores given by the patient) should be included in the results.
Author Response
Response to Reviewer 3 Comments
Point 1: In the Introduction section, the authors have explained the abbreviations (vNOTES and LESS) used many times. Usually, abbreviations are explained only once.
Response 1: Thank you very much indeed for your comments. We have accepted your suggestion and made corresponding modifications. Please see the manuscript for details.
Point 2: In the Results section the authors give the number of pregnancies obtained; there is no information on whether these were natural pregnancies or those obtained through assisted reproduction techniques; if so, which one?
Response 2: Thank you very much indeed for your comments. Maybe you can find the answer in Table 4.
Point 3: The authors still did not include information on how many points on the pain scale (NRS) patients gave after using both methods.
Response 3: Thank you very much indeed for your comments. The NRS (numerical rating scale) for pain is an 11-point scale, with 0 representing “no pain” and 10 “unbearable pain”; 1-3 points correspond to mild pain, 4-6 points to moderate pain, and 7-10 points to severe pain. Please refer to the supplement.
Point 4: If the authors compare the duration of pain, they should give at least a cut-off point.
Response 4:Thank you very much indeed for your comments. The NRS (numerical rating scale) for pain is an 11-point numeric rating scale, with 0 representing “no pain” and 10 “unbearable pain.” 1-3 points are mild pain, 4-6 points are moderate pain, and 7-10 points are severe pain.
Point 5: In my opinion, if we compare pain when using some methods, then data (in the form of NRS scores given by the patient) should be included in the results.
Response 5:Thank you very much indeed for your comments. The main limitation of this study is the lack of randomization due to its retrospective design. Further prospective studies with larger sample sizes are required to draw firmer conclusions.

Round 2
Reviewer 2 Report (Previous Reviewer 3)
M&M Section: Obstetrical data were collected by telephone interview. The authors should address recall bias as one of the drawbacks of the study in discussion.
Term “frequent unprotected sex” should be explained. Patient selection should be better explained.
Clinical pregnancy cannot be defined by was defined as amenorrhea, positive urine/blood pregnancy test. Positive urine/blood pregnancy tests represent biochemical pregnancy. This issue represents a problem for the setting of presented study. Although authors got a suggestion to properly explain, they did not provide explanation for intraoperative blood loss. Sentence “The mean blood loss during surgery was estimated by weighing and area methods.” As previously mentioned, following is not measure of surgical experience: “All surgeries were performed by the same surgical team.”. If the authors want to draw back conclusions about the importance of surgical experience for vNOTES , they are supposed to find a proper measure for surgical experience. Surgical experience is discussed on page 10.
Results Section: Parity is not presented with median in range, as it is explained in M&M. Sentence “The NRS (numerical rating scale) for pain is an 11-point scale, with 0 representing “no pain” and 10 “unbearable pain”; 1-3 points correspond to mild pain, 4-6 points to moderate pain, and 7-10 points to severe pain.” with appropriate reference belongs to M&M section. Data on postoperative infection was not mentioned in the M&M section, although authors refer to infections in the results section. Results “we find that vNOTES does not increase the risk of postoperative infection (39.02% vs. 45.86%, P>0.05).” as they have only data on postoperative fever frequency and claim that postoperative infection frequency rate was 0.
Discussion Section: Flowchart does not belong to the discussion section, as previously outlined.
References: Reference number 1 is missing. Despite previous comments, the reference list is incorrect. References 4,5,13,14,15,18,21,22,27, 29 and 32 are not properly cited. Only 45% of all cited references were published during the past five years.
Language editing should deal with the following sentence “There was no significant difference ... “ page 6. Instead “was” I would suggest “were”.
Author Response
Please see the attachment.

This manuscript is a resubmission of an earlier submission. The following is a list of the peer review reports and author responses from that submission.
Round 1
Reviewer 1 Report
The authors evaluated the efficacy and safety of transvaginal natural orifice transluminal endoscopic surgery in the treatment of female infertility. They concluded that transvaginal natural orifice transluminal endoscopic surgery provides a new, less invasive approach for infertility diagnosis and treatment, which is particularly suitable for women who have special aesthetic requirements. They also pointed out that transvaginal natural orifice transluminal endoscopic surgery is safe and feasible, which may be an ideal choice for scarless infertility surgery.
I read the study with great interest. My concerns are as follows:
1. Abstract – Before aims of the study please add a few sentences regarding the transvaginal natural orifice transluminal endoscopic surgery. Also, the methodology should be better described (inclusion criteria, outcomes of the study).
2. Please provide exact p-values in abstract, as well through the results section, instead of p<0.05 or p>0.05.
3. References in text should be presented at the end of sentence in square brackets. Please revise.
4. In the introduction the authors used the abbreviation vNOTES but they do not provide the meaning of that abbreviation. Previously only abbreviation for NOTES was described. Please revise.
5. Methodology - Authors should add the reference and date of approval by their ethics committee.
6. Design of the study - The authors should clearly present all variables that were measured/investigated in the study, also the comparison groups should be clearly described.
7. Primary and secondary outcomes of the study should clearly be identified in methodology.
8. Which statistical test was used to test normality of distribution of the data? Please update.
9. Description of surgery (both compared techniques) should be added in methodology.
10. Resolution of both flow-charts is extremely poor and should be improved.
11. Tables – All abbreviations presented in Tables should be mentioned in the legend of the Table, regardless of whether the abbreviation was previously described in manuscript. Some abbreviations were not presented, e.g. vNOTES, LESS…
12. If available, intraoperative photographs should be included.
13. Finally, the advantages of NOTES are well known and described several times in literature. This study was performed in patients treated because of female infertility, but this is not a novelty. As it was retrospective, with a limited number of the patients unfortunately I do not see much benefits for the readers.
Reviewer 2 Report
The topic of the manuscript is interesting.
I have a few comments below:
In the Introduction section, authors should explain the abbreviations: vNOTES and LESS. Full expansions of abbreviations are only in the title of the manuscript.
In the Materials and Methods section the authors used the phrase: partners…. had a normal sex life for a year without contraception….; -what does normal sex mean?; authors should define how frequent regular intercourse should be according to WHO recommendations
In the Results section the authors give the number of pregnancies obtained; there is no information on whether these were natural pregnancies or those obtained through assisted reproduction techniques; if so, which one?
The authors did not include information on how many points on the pain scale (NRS) patients gave after using both methods
In the Conclusions, the authors give:„lower complication rates than LESS”. What complications do the authors mean? According to the results obtained, there were no complications when using both methods-please clarify.
Reviewer 3 Report
I appreciate the opportunity to review the manuscript entitled “Diagnosing and Treating Infertility via Transvaginal Natural Orifice Transluminal Endoscopic Surgery versus Laparoendoscopic Single-site Surgery: A Retrospective Study” submitted to Journal of Clinical Medicine.
Female infertility represents a major health burden worldwide, causing significant morbidity and impairment of HRQoL. In the era of promoting minimally invasive treatments, the subject of the presented manuscript is both important and very interesting in modern gynecological practice.
The authors rightfully underlined the importance of new surgical approaches in infertility evaluation and treatment. Hence, they defined the study objective “to demonstrate the feasibility, safety and efficacy of vNOTES in the surgical treatment of female infertility”. Still, data on infertility treatment in the presented study is missing. Moreover, the presented manuscript has a lot of methodological flaws and drawbacks.
Despite publications on NOTES surgery in infertility treatment are lacking, which makes research in this area especially important for the clinical practice, presented work has numerous important points missing, out of which the main issues represent inadequate methodology and questionable data on ethics.
Some Specific Comments:
1. Abstract: The study did not include a total of 193 patients (Pg 1), but 174 (Pg 2). Pregnancy outcomes were not really analyzed, as only data on CPR were statistically evaluated and compared. Finally, the main issue of vNOTES use in infertility treatment in modern clinical practice is not its undisputable cosmetic superiority over LESS.
2. Throughout the manuscript, there are spelling and grammar errors.
3. In the Introduction section, the authors cited a total of 19 references which represent 45.24% of all references used in the manuscript.
4. In the Materials and Methods section, authors describe their study as retrospective, conducted from September 2017 to December 2021, while the deadline for postoperative follow-up was June 2022. Authors claim that all patients gave their informed consent for the study (page 8). This issue on study design is unclear. When did the patients give the informed consent for the study if it is retrospective? If the study was cross sectional, then patients could have provided their informed consent. A total of 193 women were analyzed and 174 included, while in abstract authors wrote that 193 patients were included. “Normal sex life” referred as an inclusion criterion is not a common term. Inclusion and exclusion criteria are not justifiable (if the age at surgery was up to 45 yrs. and patients were followed up for at least 1.5 yrs.; then the patient who was 45 yrs. at the time of surgery was 46.5 yrs. old at the end of follow-up period, and the pregnancy rate in this age is lower than in younger woman). There are no data on the following variables presented in the Results section: which method was used to estimate mean blood loss during the surgery, which criterion was used to define postoperative fever, what is NRS and how it was used, etc. Statistical methodology is also questionable: some data are expressed as median and range, and parametric statistical methodology was used. There is no data on how the authors obtain p values for comparisons of data expressed as median and range. Moreover, some data provided as mean±SD doesn’t seem right: i.e. mean blood loss in Table 2 with mean of 29.87 and SD 34.27. SD are also quite high for operation duration (˃30% of the mean value). Perhaps these data require presentation as median and range, and comparison with non-parametric tests, but it is up to Journal’s statistical Editor to provide final judgement on statistical methodology and presentation of the results in the tables. Following sentence requires explanation:” We used clinical pregnancy as the primary endpoint, which is defined as time from surgery to the presence of at least one gestational sac on ultrasound.” It is properly explained if the author evaluated occurrence of pregnancy or the time to achieve it as study primary endpoint. Other endpoints are not mentioned in M&M.
5. Results section: Flowchart 1. is unnecessary. Sentence “Demographic data, operation records, and pregnancy outcomes were collected belongs to the M&M section. Why did the authors use IQR instead of range in Table 1? There is no clear data on further treatment of infertility, i.e., how the patients conceive? Did they use ART in all patients? Figure 2: why the authors did not compare frequencies of full-term pregnancies in the groups? If the procedures were simple diagnostic, why did the patients require a mean postoperative hospital stay of more than 3 days and why did the operations last almost 1.5 hours? If the procedures were therapeutic, why there are no data on operative interventions performed?
6. Discussion section: Authors cannot discuss vNOTES in line of infertility treatment as data on infertility treatment are missing in the results. What is the meaning of the construction “which indicating that vNOTES did not increase the difficulty and patients exposed to anesthesia”? Authors cannot discuss postoperative infections if they presented only frequency of postoperative fever and did not even explain the definition of fever they used. Data on postoperative infections are missing in the manuscript: i.e., microbiology, CRP, leucocyte count, surgical site infections, etc. It is contradictory to state that “vNOTES did not increase the risk of postoperative infection (39.02% vs. 45.86%, P˃0.05). In this study, no infectious adverse events were found in any patients.”-page 5. Meaning of the text “vNOTES was a feasible and safe surgical technique for performing infertility”-page 6. The presented results do not data on surgical experience of the surgeons who performed evaluated procedures, therefore, authors cannot draw conclusions about benefits of vNOTES in the hands of experienced surgeons.
7. Conclusion section: Last sentence is not understandable.
8. References are outdated. Out of a total of 42 citing publications only 10 are up to date (published from 2018-present). A total of 5 publications are related to general surgery, while 4 are related to female sterilization procedures. and less than 50% (9 out of 23) were published over the past 10 years. Almost half of the references (45%) were used to support claims made in the Introduction section.
9. The citation style seems inadequate, i.e., ref #4, ref #5, ref 13, ref#22, ref #33, etc.